# Revisiting Nature’s “Unifying Patterns”: A Biological Appraisal

**DOI:** 10.3390/biomimetics8040362

**Published:** 2023-08-13

**Authors:** Guillaume Lecointre, Annabelle Aish, Nadia Améziane, Tarik Chekchak, Christophe Goupil, Philippe Grandcolas, Julian F. V. Vincent, Jian-Sheng Sun

**Affiliations:** 1Institut de Systématique, Évolution et Biodiversité, UMR ISYEB 7205 CNRS MNHN SU EPHE UA, Muséum National d’Histoire Naturelle, CP 50, 45 Rue Buffon, 75005 Paris, France; 2Bioinspire-Museum, Direction Générale Déléguée à la Recherche, l’Expertise, la Valorisation et l’Enseignement (DGD REVE), Muséum National d’Histoire Naturelle, 57 Rue Cuvier, CP 17, 75005 Paris, France; 3Institut des Futurs Souhaitables, 127 Avenue Ledru Rollin, 75011 Paris, France; 4Laboratoire Interdisciplinaire des Énergies de Demain (LIED), UMR 8236 CNRS, Université Paris-Cité, 75013 Paris, France; 5Nature Inspired Manufacturing Centre, School of Engineering, Heriot-Watt University, Edinburgh EH14 4AS, UK; 6Structure et Instabilité des Génomes, UMR 7196—U1154, MNHN CNRS INSERM, Muséum National d’Histoire Naturelle, 43 Rue Cuvier, 75005 Paris, France

**Keywords:** bioinspiration, biomimetics, biomimicry, unifying patterns of nature, biology, principles, evolutionary theory

## Abstract

Effective bioinspiration requires dialogue between designers and biologists, and this dialogue must be rooted in a shared scientific understanding of living systems. To support learning from “nature’s overarching design lessons” the Biomimicry Institute has produced ten “Unifying Patterns of Nature”. These patterns have been developed to engage with those interested in finding biologically inspired solutions to human challenges. Yet, although well-intentioned and appealing, they are likely to dishearten biologists. The aim of this paper is to identify why and propose alternative principles based on evolutionary theory.

## 1. Introduction

Bioinspired or biomimetic innovation depends on effective communication between many disciplines, with biology being central to its success. In order to maximise the chances of fruitful discourse, the basics of biology should be understood by those involved in the bioinspiration process. Biologists have an essential role to play in facilitating this comprehension [1].

The Biomimicry Institute, an American non-profit organisation that seeks to promote biomimicry through communication and networking, has produced ten “Unifying Patterns of Nature” [2] (five of which were already published by J. Benyus in 1998 [3]) to support learning about living systems. As the Biomimicry Institute explains in relation to these “nature’s patterns”, “Our intent is not to present this as a definitive and exhaustive list. Rather, it is a work in progress that we hope will be informed and enhanced by the growing community of biomimics who are practicing applying nature’s lessons to their designs”.

In the spirit of joining this work in progress, our article addresses each of the ten “nature’s unifying patterns” in light of theories from biology and physics, and discusses ways in which they might be reformulated along more scientific lines (Table 1).

## 2. When “Nature” Is the Subject of the Verb

Our first observation relates not to a “Unifying Pattern of Nature”, but to the way in which the ten “patterns” are presented by the Biomimicry Institute. “Nature” is systematically employed as the subject of sentences (“nature uses”, “nature provides”, “nature builds” etc.). This phrasing is problematic because, in science, nature is not a casual factor or active agent with its own inherent self-determination. Rather, it is a series of phenomena, a large system of varied interactions without any individuality, that require interpretation through the process of scientific enquiry. If this were not the case, the entire scientific process would be inconsequential: nature would be considered “omnipotent” on principle and therefore devoid of the need for explanation.

Human beings can be susceptible to thinking that there is agency or intention within or beyond nature, a cognitive bias called “hypersensitive agency detection” [4,5] or “intentionality bias” [5,6]. Since the Scientific Revolution of the 17th and 18th centuries, the role of science (as stipulated by the philosophers and scientists of the time, such as Diderot and Buffon) has been to overcome this cognitive bias and explain nature by examining its properties. From then on, “providence” had no place in science. Single, overarching explanations based on supernatural or divine causes (which could not be tested experimentally) were excluded as flawed. Instead, natural phenomena had to be evaluated logically and attributed to specific, tangible causes via rational explanation.

By using nature as a subject in statements about patterns in biology, a “universal provider” or “global causal agent” appears, once again, to be attributed a deterministic role. “God” is simply being replaced by “Nature”. This attribution may be involuntary and solely based on the Biomimicry Institute’s desire to communicate with audiences with non-scientific backgrounds. However, the risk is that this wording perpetuates inaccurate assumptions about the living world.

Furthermore, using nature as the subject of sentences reinforces its interpretation as everything that humans themselves have not created, i.e., a concept in opposition to humankind. This “nature-otherness” [7] is not only egocentric but also presents several challenges: firstly, it suggests that nature can be used as an (inexhaustible) “external” resource with no consequence to humans; secondly, it may encourage human beings to dispute their biological origins, as they are led to assume they are distinct from nature.

Paradoxically, although nature may be erroneously presented as separate from humankind, this does not prevent humans from assigning their *own* personality traits to nature. Nature is sometimes described as “parsimonious”, “frugal”, “intelligent” or even “rational”, indicating that human beings attribute their own characteristics to the world around them in order to understand both it and themselves. Statements such as nature is “logical” or “rational” (for example, in the case of François Jacob’s famous book “*La logique du vivant*” [8]) are similarly erroneous. Life is not inherently logical. Logic is an epistemological and mental trait associated with cognition (the process of acquiring knowledge and understanding). Thus, in scientific investigation, logic is an observer’s characteristic, not a feature of the “natural” phenomena we seek to understand. One could object that “logic” can be observed in animal behaviour when they change or develop their interactions with their environments. Yet, while these properties do have effects on some generations in some lineages, this does not justify qualifying “Nature” or “Life” as a whole as “logical”.

Confusion between the properties of the observer (us) and the properties of what is observed (nature) often occurs when communicating about the biological world in order to generate empathy for “nature”. Yet, by saying that “nature is x”, or even by rejecting “x” (because evolution would have no reason to be “x” (as described by Anderson [9], in relation to parsimony)) we are deluding ourselves). It is we, as observers, who create these hypotheses; we are simply projecting human behaviours or properties onto external phenomena, a classic bias called “apparent behaviour” [10].

The same applies to human intentions versus the natural world’s inherent lack of intention. When we say that there are “natural” strategies, designs, programmes, or plans, we are creating metaphors that, despite being superficially useful as a means of communication, create intellectual obstacles further down the line [11,12,13,14,15]. These anthropocentric notions ultimately limit our ability to understand the living world by way of its *own* processes and properties, in line with the modern theory of Biology.

What do these metaphors have in common? They share two attributes. Firstly, *anticipation*: they assume that the result of a given process is *expected* by the living world. As such, they are teleological notions, i.e., explained in terms of the *outcome* rather than the cause through which they arise. Secondly, *causation*: they use “order” found in nature as an indication of *purposefulness*. Human beings are designers, creators, and builders; as such, we mentally anticipate the functional properties of the object we are making and adapt the construction process to generate these properties. That is to say, we begin with the end in mind. It is the order enshrined in the human process of creation that delivers the desired result, i.e., the alignment of form and function.

Yet, while humans design structures purposefully, “nature” does the exact opposite. Behind the regularities of the living world lies “disorder” in terms of spontaneous, ongoing variation on a vast scale. Populations of organisms vary at all levels of biological organisation: each individual organism is unique; even siblings are not identical. This tremendous biological variation is then “blindly” shaped by local constraints (physical, chemical, or biological) through natural selection. Most organisms do not survive this process of natural selection. This explains why only a small number of the eggs or seeds produced by any biological generation will reach their adult stage and why just a fraction of these adults will go on to have offspring.

We are often oblivious to the huge number of individuals that have died, relegated to an invisible “evolutionary graveyard”; instead, we simply observe those organisms that thrive today (a counter-intuitive but logical explanation stated by Maupertuis in 1754 [16]). These organisms “succeeded” because the forms they embodied were, just by chance, compatible with local survival. Their forms were generated at random via genetic mutation (and other kinds of changes). It was their (slightly) better *functionality* that then acted as a filter by way of natural selection. This is because, in the living world, *function* (i.e., what a biological entity does and how well it does it) maintains form and drives its evolutionary changes. Note also that the resulting forms have to be incorporated into an existing network of biological trade-offs.

This is not the way in which humans reason when we design objects. Human beings think purposefully and use shape to achieve desired functions. We also tend to minimise the number of design iterations necessary to achieve our goal, and seek to create the best result as quickly as possible. As such, most of us are reluctant to accept that the congruence we observe between form and function in nature was never anticipated. We inevitably transfer our “cognitive reflexes” to the living world, unwittingly believing that “nature” preemptively “designs” shapes to optimise particular functions, as we do.

Bioinspiration, a domain that exists at the intersection of biology and design, is particularly susceptible to these ambiguities. Yet, it also holds the potential to improve biological understanding amongst a wide range of stakeholders. We believe this comprehension is fundamental to bioinspiration’s long-term success: bioinspired designers and engineers can only benefit from understanding evolution and the key principles of biology as a science. By truly grasping how nature operates, flaws and miscalculations are more likely to be avoided in the development of bioinspired processes, products, and systems.

## 3. What Is a Pattern?

Another aspect that could cause confusion amongst the Biomimicry Institute’s audience is the use of the term “pattern” to describe universal arrangements found in the natural world. The Biomimicry Institute employs the term pattern in the sense of “principles that guide biological systems’ functioning”. Unifying concepts, such as evolutionary theory, do, of course, exist in biology and are fundamental to its understanding (as explained in this article). However, the word pattern is more commonly taken to mean “regularity in a combination of structures” or the way in which a formation is repeated in a regular way. Thus, the reference to “patterns” in the natural world can lead us to visualise a zebra’s stripes, the hexagons of a honeycomb, or even features (such as scales) present in multiple species.

Patterns (in the sense of *regularities*) found in nature are underpinned not just by biology, but also by mathematical and physical principles. This is the case for both nature’s living and non-living elements; for example, one might observe the similarity between the branching formation of a river and the branching veins of a leaf. These natural regularities were first brought to public attention by D’Arcy Wentworth Thompson in 1917 [17] and subsequently expanded upon, notably by Ball in 2016 [18]. As Ball explains, physical and mathematical rules that create self-organisation appear to both *“restrict the options for adaptive change and to offer new adaptive opportunities”.* Thus, they do not replace biological principles, but rather work alongside them; the resulting shapes (fractals, spirals, waves, etc.) are a complex medley of math, physics, chemistry, and biology operating in parallel.

A second use of the term “patterns”, this time in evolutionary thinking, refers to the distribution of organisms’ characteristics across biodiversity [19,20,21]. Here, the idea of regularity is used in the sense of a given structure being repeatedly found in combination with others across taxa. For instance, in current biodiversity, vertebrae are found in all vertebrates, not beyond them. Some extant vertebrates have feathers (i.e., birds). However, no animal has feathers without also having vertebrae. Such patterns have a phylogenetic explanation: ancestors in which feathers appeared are more recent than ancestors in which vertebrae appeared; and feathers appeared uniquely among vertebrates. Furthermore, all animals with a pygostyle (a fused set of bones at the posterior end of the vertebral column) instead of a tail also have feathers, and all of them are vertebrates as well. Phylogenetic analysis seeks to uncover “a common pattern of inter-nested attributes” [22]; in other words, to identify organismal features that are repeatedly combined in an inter-nested way based on species’ genealogy.

The disparity between a “guiding principle *of* nature” vs. a “regularity *in* nature” may be confusing to bioinspired designers. We suggest that the Biomimicry Institute’s “Nature’s Unifying Patterns” be reworded to refer to “Principles of the living world” in order to avoid any misunderstanding between these two interpretations.

It is in this spirit that we examine the Biomimicry Institute’s ten “unifying patterns” of nature, highlighting why these principles are not consistently in line with a scientific understanding of the biological world. We argue that bioinspiration does not need to personify nature to be useful; rather, we advocate for an informed bioinspiration that recognises the evolutionary processes and physical forces that brought about our “biological muses”. We suggest that these “unifying patterns” be adjusted to take evolutionary theory into account while retaining their straightforward, communicative style. Finally, we put forward eight additional biological principles for enlightened bioinspiration.


**Pattern 1. “Nature uses only the energy it needs and relies on freely available energy”**



*“Energy is an expensive resource for all organisms; the risk of using excess energy is death or the failure to reproduce. Therefore, they use it sparingly, tailoring their needs to the limited amount of energy available. While “no energy is free”, because all energy requires expenditure of energy to obtain it, nature’s sources for energy are freely available because they are renewable, are found locally, and don’t need to be mined. Freely available energy includes sources such as electrons from sunlight used by plants for photosynthesis, rising air currents, wind, dissolved minerals from deep sea vents, decomposing organic materials, and nutrients from plants and animals that organisms feed upon. Two major energy expenditures for organisms are obtaining the energy (e.g., through photosynthesis or finding and capturing food) and growing materials that make up their bodies and homes. Organisms use low-energy processes to reduce the amount of energy they need. Those processes usually involve self-assembly, building from the bottom-up (small elements to large), using modular or nested structures, building at ambient temperatures and pressures, and making use of multi-functional design”.*


The statement “*Nature uses only the energy it needs*” seems to imply that this is a choice a biological system has made. Rather, the opposite is true: the “energy sobriety” sometimes found in living organisms and ecosystems is a consequence of limited energy availability. It is not possible for any system, and living organisms are no exception, to consume more energy than is available. This is because, in physics, energy is a “conserved quantity”, indicating that it cannot be created, nor even destroyed, but only exchanged or converted into another form (Emmy Noether’s theorem (1918) and Helmholtz’s formulation (1850)).

Within an ecosystem, there are two main ways of acquiring energy: autotrophic and heterotrophic. The autotrophs, or primary producers, use energy either from light (photosynthesis), or from inorganic chemical reactions (chemosynthesis) to make their “food”. In both cases, they convert an abiotic source of energy in their environment into biological components. The heterotrophs cannot produce their own food and therefore obtain energy by consuming other organisms within a trophic network. The higher an organism is in a specific food network (e.g., apex predators), the lower the energy conversion rate from one level to the next.

At an ecosystem level, this poor “return on investment” at each trophic level is usually balanced through the relative abundance of primary producers and the “low cost” of their energy harvesting/conversion structures (such as leaves and chloroplasts). However, as the Biomimicry Institute correctly states, ‘*no energy is “free” because all energy requires expenditure of energy to obtain it*’. Even the flow of electrons from the sun requires extraction via the presence of photosynthetic capture and conversion surfaces in plants. As with any transformation of matter, these living structures represent an energy cost for construction and maintenance. There is no such thing as a free lunch! The notion of “*freely available energy*” exploited by nature is therefore rather vague as part of this “pattern of nature”, (although we understand its intention to contrast with humans’ mining of remote fossil fuels for energy consumption).

The level of energy conversion within biological systems also varies according to the level of organisation (i.e., cell, organ, organism, population, etc.) in question. While there are many examples of biological systems at all scales exhibiting apparent low levels of energy conversion (“minimising their energy use”), there are also counter-examples that suggest that this “pattern” is not absolute in nature and therefore should not be considered a rule.

For example, at the *population level*, most species exhibit high levels of energy conversion (“energy expenditure”) producing far more eggs or seeds than the local environment could ever support. For instance, most teleostean fishes lay several hundred thousand eggs, with the maximum reached by the sunfish (300 million eggs), most of which die prematurely. Not all eggs or seeds become adult organisms, whatever the embodied energy in each, either because of abnormal development or because of predation. This is one of the key tenets of Darwin’s reasoning: random variations within growing populations of organisms are “filtered” by environmental parameters, both biotic and abiotic. Organisms respond to this by expending considerable amounts of energy in order to counterbalance the inevitable losses of eggs and juveniles.

The energy “expenditure” involved in reproduction is not always obvious, particularly in terms of individuals lost because of abnormal developments. These abnormalities occur because development is not so much the unfolding of a genetic “programme” (as we might imagine [23]), but rather a process of construction in and of itself [24]. In a large number of cases, reproductive development fails. Even in human populations, the proportion of pregnancies that will lead to a newborn baby is surprisingly low: natural human embryo mortality ranges from 40% to 60% [25]. Once again, each generation (human or otherwise) uses much more energy in reproduction than their adult population appears to require.

Do species expend an *appropriate* amount of energy to ensure the production of the next generation, given high levels of predation and abnormal development? Unfortunately, no objective test can ascertain this: there is no experimental design to evaluate whether sunfish females could lay more eggs or not. In any case, behind this *apparent* “optimisation” lies the invisible demise of millions of individuals. Once again, the statement “*Nature uses only the energy it needs*” is problematic because it implies some sort of conscious target within living systems.

Furthermore, deciding whether “nature” is wasting or saving energy depends on what we are evaluating: evolutionary processes or their results. At the scale of egg production, energy is undoubtedly “lost” for the species in question, but organisms that thrive (via natural selection) are those that (to some extent) minimise energy conversion. In other words, evolutionary processes “demand” an extravagant expenditure of energy but result in biological systems that use the least energy possible (from a physiological perspective). In this sense, the Biomimicry Institute is partially correct in stating that “*Organisms use low-energy processes to reduce the amount of energy they need*”. However, even at the organism level, there are examples where energy is converted into developing forms that appear to serve little purpose or even lead to “unnecessary” energy expenditure. For example, the human tail develops nine vertebrae, of which four degenerate in the womb; the five others become the vestigial coccyx. Similarly, the giraffe’s left recurrent laryngeal nerve makes a “useless” five-meter-long circuit to reach the larynx, a seemingly extraordinary investment of energy for little gain. The same circuit was estimated to be 28 m long in the sauropod *Supersaurus* [26].

These structures would never have been “designed” with efficiency in mind; they result simply from the fact that each organism carries within them the evolutionary stages through which their ancestors passed. This “evolutionary burden” or “phylogenetic heritage” cannot be avoided within the living world and needs to be understood if biological systems are to be effectively transposed to bioinspired innovation. This is because understanding a biological system in its entirety can lead to different decisions being taken during the design process. Certain structures may need to be retained (to ensure the efficient functioning of the bioinspired product or system); others may need to be abandoned as they do not serve a purpose in a human-designed output.

**Pattern 1 could be reformulated as:** In terms of reproduction, both internal and external constraints can lead to higher energy expenditure than might be inferred from observation of adult populations. Evolutionary processes require considerable amounts of energy—either from the organism’s point of view (number of gametes produced) or from the population’s (number of deaths). However, natural selection seems to ultimately favour physiological systems that minimise energy expenditure.


**Pattern 2. “Nature recycles all materials”**


“*In nature, one organism’s waste or decomposing body becomes a source of food and materials for other organisms. While we talk about “recycling,” “upcycling” is a more accurate description of what happens in nature. There are usually many organisms, or more accurately, ecosystems of organisms, that break down complex organic materials and molecules into smaller molecules that can then be taken up and reassembled into completely new materials. Just as there is a hydrological cycle, there are many other cycles involving organic matter (carbon cycle, nitrogen cycle, etc.) that function as local, regional, and whole-earth systems*”.

This statement could be considered legitimate if we consider “nature” to be all organic and inorganic matter over indefinite timescales (following Maris’ vision of “nature-totality” [7]). Yet if nature is defined as *living organisms* (and therefore only part of the material world), its capacity to recycle has certain limitations. Indeed, the idea of “no waste” in nature has already been identified as a misconception by Shyam et al. [27]. Here, we present five examples of the limits of biological recycling.

During the “Great Oxygenation Event” 2.4 billion years ago, cyanobacteria proliferated, producing huge quantities of oxygen that reached toxic levels for multiple organisms. This led to the extinction of many anaerobic species at the time. “Nature” was not able to recycle one of its outputs, and it took millions of years for living organisms to adapt and benefit from this modified environment. Similarly, the super greenhouse climate of the early Triassic was driven by vegetation collapse after the volcanic events of the Permian–Triassic Mass Extinction: plants did not sequester organic carbon, and carbon dioxide was not “recycled”. This led to increased levels of greenhouse gases and a prolonged period of extremely high surface temperatures, which delayed biotic recovery [28]. Furthermore, during the Carboniferous and Permian periods, hard plant tissues were not “recycled” and accumulated in the form of oil and coal. In a similar vein, today’s global warming is a consequence of one species burning oil and coal (fossilised organic matter) and creating greenhouse gases; “nature” cannot recycle carbon dioxide within the necessary time frame to avert global change. Last but not least, no living organism can recycle chalk that has accumulated for millions of years in sedimentary layers hundreds of metres thick.

Ultimately, “recycling” is a matter of time scale. Earth’s living systems cycles are generally much longer than those that characterise human activities. Since the Industrial Revolution, access to stored energy in the form of fossil fuels has allowed humanity to develop infrastructure and products faster than ever before. However, nature’s capacity to recycle the “waste” associated with this modernization (i.e., its decomposition into molecules then assimilated into new organisms) is not in line with humanity’s timescales, threatening the resilience of ecosystems on which humans depend. Remembering this variable of time and recognising the finite character of our planet are two sides of the same coin.

Interestingly, a different kind of “recycling” occurs during the course of evolution: not of matter but of shape. A biological structure can gain supplementary functions without changing its form. For instance, human beings’ opposable thumbs appeared at the origin of primates (arboreal mammals) more than 60 million years ago. These opposable thumbs were used to grasp branches (similar to claws that appeared in other arboreal vertebrates like chameleons, several lineages of birds, etc.). Today we use these opposable thumbs for hitchhiking and for indicating agreement/well-being, a new signalling function. The same phenomenon occurred with feathers. Feathers first appeared in dinosaurs, where their primary functions were insulation as well as body protection and display. Over time, their presence gave certain species an evolutionary advantage (possibly allowing them to escape predators more quickly), and feathers used for flight appeared [29].

Different parts of biological features can also be “recycled” to fulfil a variety of functions. For instance, in Antarctic teleostean fishes (notothenioids), the pancreatic trypsinogen-like serine protease gene is used by the liver to deliver antifreeze proteins to the blood. Similarly, in eelpouts, the sialic acid synthase gene is employed for adaptation to cold Antarctic waters [30]. This ability to recycle or transform a particular structure’s function could be of particular interest to bioinspired designers looking to address questions of multi-functionality and the development of systems responsive to change.

**Pattern 2 could be reformulated as:** The living world has an extraordinary (but not infallible) capacity to recycle organic material. In any given ecosystem, a diversity of organisms reuse, scavenge, or decompose matter into components taken up by other forms of life. However, “recycling” can take millions of years, and some organic materials have never been “recycled” at all.


**Pattern 3. “Nature is resilient to disturbances”**


“*Being resilient is about having the ability to recover after disturbances or significant, unpredictable changes in the local environment, such as those caused by a fire, flood, blizzard, or injury. Diversity, redundancy, decentralization, self-renewal, and self-repair can all enable resiliency in nature and the ability to maintain function despite a disturbance. At a systems level, “diversity” refers to the presence of multiple forms, processes, or systems that meet a functional need. Diversity can include a variety of behavioral, physical, or physiological responses to a change in the environment. “Redundancy” means that there’s more than one representative system, organism, or species that provides each function, and that there’s overlap so the loss of or decline in one representative doesn’t destroy the whole system. “Decentralization” means that the mechanisms maintaining those functions are scattered throughout the system, not located exclusively together, so that a localized disturbance doesn’t remove one or more vital parts of the whole system. “Self-renewal” and “self-repair” are terms that are more often applied at the cellular or organismal level, but self-renewal can also be applied in ecological contexts. For the former, the terms mean that organisms have the capacity to generate new cells, heal wounds and damaged organs, respond to bacterial and viral threats, and more*”.

These statements hold true in many contexts. Biodiverse, healthy ecosystems are more resilient to disease, predation, or environmental change than a monoculture, for example. This is because functional redundancy has probably been favoured (through natural selection) as a form of resistance to structural alterations at several biological scales [31]. The diversity of interactions between different species is also a well-known factor determining ecosystem resilience. Large ecological networks reach a kind of “self-regulation” whereby an increase in one species’ abundance decreases its per-capita growth rate. This self-regulation is delivered through several aspects of the functioning of food webs, as described by Barabas et al. [32]. Furthermore, the decentralisation of functions has been shown to provide resilience in social insects, an approach that has inspired the management of human infrastructure [33].

In terms of self-repair, the ability to heal is a characteristic of all multicellular organisms. There are many different self-repair mechanisms in the living world, for example, lianas being able to seal lesions quickly to maintain functional integrity [34] (even inspiring biomimetic self-repairing materials [35]). Even more spectacular is the regeneration found in some metazoans, such as starfish regenerating lost arms.

Yet these mechanisms of functional redundancy, decentralisation, density of interactions and self-repair are not necessarily all found at the same level of biological organisation. Healing has no equivalent in ecosystems unless the term is used metaphorically. Functional redundancy can be found at a molecular level within animals, but generally not at the anatomical level. Self-regulation within a food web is very different to the physiological regulation of an organism.

Crucially, all these mechanisms confer resilience on living systems *only up to a certain point*. As it stands, this pattern’s wording could be misleading, suggesting resilience is an absolute property of nature. Yet, as already discussed, most eggs, embryos, and juveniles die before reaching adult stages. Species become extinct: the world has seen many mass extinction events prior to the Anthropocene, with the five main ones being called “the big five”. Ecosystems can be transformed through ecological successions, whereby resilience is a dynamic process and the “identity” of the ecosystem depends on time and social perceptions. Indeed, anthropogenic disturbance can maintain ecosystems in a specific state of conservation value from a human perspective (such as grassland ecosystems, regulated by grazing livestock to prevent their evolution into forests). Ecosystems also collapse, sometimes irreversibly. For example, faced with chronic anthropogenic pressures, tropical reefs change from coral-dominant to macro-algae-dominant systems, making recovery difficult, if not impossible. Likewise, scientific models predict large-scale ecosystem collapse this century: the Amazon basin, Congo Basin, and Gobi Desert by 2050, and the Coral Triangle by 2060 [36] due to unprecedented human pressure.

These issues are of particular relevance to the bioinspiration domain because of its capacity to contribute to regenerative development and biodiversity conservation. While life as a whole may not be fragile, the ecosystem services on which humanity depends are increasingly so. Bioinspired designers need to understand biological resilience in order to effectively transpose this quality to human systems and technologies. Reasons and timescales for resilience are too diverse to be reduced to a simple property of “nature”. Rather, the designer needs to identify the dominant factor(s) providing resilience in a particular context (i.e., at a specific biological level) as well as the time periods over which it/they operate(s). Only then can they evaluate whether transposing a biological model to a human system is likely to provide the expected resilience. Additionally, this should be performed with the acknowledgement that no living system is infallible!

Finally, resilience is neither a choice nor a designed property but the necessary result of an evolutionary trade-off, which itself governs most developments in the living world. “Good enough’ seems to be the hallmark of evolution [37], although this is still debated. Being ‘good enough’ is the result of either neutrality or trade-offs, which themselves are the outcome of biotic and abiotic constraints (driven by the laws of physics). Biological resilience is the result of the interplay of these constraints, which collectively define the boundaries within which biological expression can take place. Trade-offs are examined in more detail below (Pattern 4).

**Pattern 3 could be reformulated as:** Ecosystems and biological entities are resilient to disturbances within certain limits. At the ecosystem level, once certain disturbance thresholds are crossed, the “identity” of the ecosystem may be changed irreversibly.


**Pattern 4. “Nature tends to optimise rather than maximise”**


“*Because energy and materials are so precious, nature seeks a balance between resources taken in and resources expended. Energy spent on excess growth, for example, could result in insufficient energy reserves or characteristics that harm an organism’s ability to survive and reproduce, which means that it won’t be able to pass on its genes. There are checks and balances in both organisms and ecosystems, some of which occur over generations. Growth for growth’s sake will result in harmful side effects. Sometimes these side effects are immediately apparent and possibly reversible, and sometimes they remain hidden for a long time until reversal is too late*”.

This pattern, as written by the Biomimicry Institute, uses vocabulary typically associated with the field of systems engineering, a domain that studies the design, integration, and management of complex human-made systems. The wording is not neutral but rather anthropomorphic, conferring the principles of non-living systems onto natural systems.

Living systems do not “maximise”: the “maximum power principle” proposed by Odum and Pinkerton [38], based on the application of electronic and thermodynamic laws to ecology, has been refuted. Similarly, it is no more correct to assert that living systems ‘optimise’, as already pointed out by Shyam et al. ([27], p. 5). Both of these statements imply the existence of an underlying principle that would directly “drive” the trajectory of a biological system, one that mirrors the thermodynamics of isolated systems moving towards equilibrium. However, the living systems that make up what we call “nature” are not isolated systems, and neither are they in equilibrium. Living organisms are open, non-equilibrium, and dissipative systems that continuously exchange energy and matter with their environment [39,40]. There is therefore no overarching principle that determines energy dynamics at the population or organismal level [41,42].

That being said, checks and balances do seem to occur in nature. Yet, these are regulated more by trade-offs than optimisations at both population and organismal scales [43], generating scenarios akin to “best under the circumstances”.

At a *population* level, according to evolutionary reasoning, species maintain their lineage through time because they tend to produce a large number of offspring. Most species multiply constantly and seem to produce as many offspring as possible, leading to an apparent “maximisation”. We often forget this because of death rates induced by (i) resource limitations, (ii) competition, and (iii) predation/parasitism or because of migration. However, populations of organisms can proliferate uncontrollably under some circumstances. One example of this is the introduction of non-native species into new territories. Rabbit introductions in Australia or in the Kerguelen Islands are well-known case studies, and similar events are increasingly common internationally. So-called “invasive species” can cause significant ecological damage, such as the northern Pacific seastar in Tasmania or the geoplanid flatworms from New Zealand and South America invading Europe.

At the *species* level, the notion of *trade-off* is also key. Every organism has a range of physical and biochemical mechanisms that can be brought into play when necessary: that is how they survive. They may even shut down completely until better conditions prevail. Yet no species is perfectly adapted in relation *to a single* structure or function; rather, natural selection generates a series of biological outcomes that can counteract each other. Indeed, a system, whatever it may be, cannot be both adapted to a particular function and to a multitude of functions. Living organisms are no exception to this thermodynamic law; their overall adaptability depends on the constitution of their systems into adapted subsystems.

There are thus trade-offs associated with finding a balance between the different functions necessary for survival. In evolutionary terms, the increased fitness (or function) of one trait is often reached at the expense of another. Within an individual, any two traits depend on resources from a common, limited pool: investment in one area means less investment elsewhere. Following this model, even the apparent “maximisation” of offspring mentioned above is part of a trade-off. For example, the larger a male howler monkey’s vocal organ (and therefore the louder their roar to attract mates and intimidate rivals), the smaller their testes and the less sperm they can produce [44]. Moreover, the ability to attract and engage with a mate by being highly visible (or loud) can hinder evading one’s predators. As a result, sexual selection can come at a price. In certain cases, even mating itself directly exposes individuals to danger, as in spiders and electric fish. Another example at a group level: most animal populations display a trade-off between competition (seeking the same resources) and cooperation (working together can help obtain more resources, such as via collective hunts).

Why is this relevant to bioinspiration? Because human innovation is often underpinned by the same trade-offs that are found in the natural world. This can provide important lessons in terms of how maximising one function (e.g., speed) can diminish another (e.g., accuracy). Even so, human innovation is not necessarily bound to the same constraints as living systems. Bioinspired products and processes can be designed with the optimisation of a single function in mind, often without the need to manage trade-offs across a range of functions required for survival or being obliged to carry the “phylogenetic heritage” of forebearers (see Pattern 1). Furthermore, bioinspired design can pull different ideas from a range of biological systems at varying scales, creating a mosaic “solution” that does not have to resemble a single living entity.

**Pattern 4 could be reformulated as:** Living systems are the result of trade-offs, not optimisation. Populations seem to ‘maximise’ reproduction and offspring, which are later filtered by environmental constraints (biotic and abiotic). Apparent optimisations in terms of species’ physical and behavioural traits would be more accurately described as being the ‘best under the circumstances’.


**Pattern 5. “Nature provides mutual benefits”**


“*Among the variety of ways that organisms interact with each other, there are many examples of interactions that provide mutual benefits. The benefits may be simple byproducts of specific behaviors—for example, when one organism’s waste is another organism’s resource—or they may arise out of close relationships that evolved over time. Mutualistic symbioses are one example of a close relationship between different kinds of organisms, where all the partners benefit from the relationship. Another kind of close relationship includes cooperation among members of a family group. Even interactions that normally harm an organism, like predation or parasitism, can include benefits when viewed at a different level. For instance, a male praying mantis might be eaten by his female mate after mating, providing beneficial nutrition to the female that will eventually bear his offspring*”.

It is correct to say that there are a multitude of ways in which organisms interact with each other. It is also true to say that natural selection can lead to different forms of cooperation. Nowak [45] identified five mechanisms that override the competition typically associated with natural selection: kin selection, direct reciprocity, indirect reciprocity, network reciprocity, and group selection. These mechanisms foster the cooperation found at multiple levels of biological organisation.

Mutualistic symbiosis or mutualism (where two or more species benefit from an ecological interaction) explains most of the very spectacular “innovations” in living systems. Examples include symbiogenesis, which explains the origin of eukaryotic cells from prokaryotic organisms (but not only [46]), or the behaviour of eusocial insects (such as ants, bees, and termites). In this regard, cooperation appears to be a key driver in the evolution of biological complexity [45]. Nonetheless, cooperation is just one type of interrelationship in the natural world and should not be considered a rule or “unifying pattern”. In fact, mutualism is more difficult to identify than the two other—and most common—interactions among species: predation and parasitism.

Aside from (most) green plants, all organisms—even unicellular ones—are predators, and most of them are prey as well (except a very limited number of ‘top predators”). Moreover, no organism escapes parasitism. Even bacteria have parasites in the form of viruses. Viruses have been on earth for so long that they manage to exploit *all* existing species. The same is true for bacteria. Interestingly, bacteria’s long history of predation and parasitism has, in some cases, evolved to form symbioses. For example, most multicellular organisms have bacterial ecosystems in their digestive tracts that provide benefits to the host and the bacteria themselves. Yet, these relationships undoubtedly started as parasitism.

We tend to forget that all living species have parasites given the last hundred years of “sanitary paradise”, relative to the rest of human history. Some three centuries ago, human beings regularly had to deal with fleas, lice, plagues, cholera, etc. Even today, the main cause of human mortality in tropical areas are diseases caused by parasites (viruses, bacteria, unicellular eukaryotes, animals, etc.).

This does not mean that bioinspired designers are required to give equal weight to parasitism and predation in their work. Certainly, developing bioinspired innovation based around mutual benefits and collaboration is a positive approach and one to be encouraged.

**Pattern 5 could be reformulated as:** Mutually beneficial relationships are found in living systems, yet they are not necessarily more significant than predation and parasitism.


**Pattern 6. “Nature runs on information”**


“*To be attuned to their environment, organisms and ecosystems need to receive information from the environment and be able to act appropriately in response to that information. This includes sending and receiving signals to and from other organisms or even within the body of an organism. This system of send, receive, and respond has been finely tuned through millions of years of evolution. Some living systems work within narrow ranges of optimal conditions, so they need to constantly monitor their environment and respond. Others have broader ranges, but still need to be able to detect and respond when conditions are such that they approach their limits (e.g., maximum survivable temperature or oxygen availability). Using feedback loops is one way to monitor those conditions. Both negative feedback loops (those that slow down a process), and positive feedback loops (those that speed up a process) are important in natural systems*”.

Receiving, responding to, and sending signals are indeed central elements of the living world. However, care needs to be taken in how we use the word “information”. Information is essentially processed and organised data and is defined by the fact that it does not *vary* according to its means of communication. For example, the same information can be contained in an enunciated, emotive speech and its transcript, despite these mediums being very different (i.e., air vibrations vs. ink on paper).

Importing the notion of “information” into biology has been intensely debated by scientists [11,12,13,47,48,49]. One main challenge is that in biology, there is no medium of communication (physical, chemical, or behavioural) that can ensure the invariance of the “information” generated. The “information” inevitably changes according to the medium used and therefore no longer meets the definition of information. If information is what does not change, we must keep in mind that in Biology everything changes, always and at all scales.

Broadly speaking, notions referring to unchanging entities (invariants), such as information, are not considered integral to biology; rather, biologists note the random variation of living things and their properties. Of course, the field of biology itself is structured and organised: when a biological entity or process needs to be named, biologists use language conventions like character homologies, taxon names, or metaphors. They develop terms that they then associate with concepts in a nominalistic way in order to designate sets of varying material entities in the living world. However, unlike during the time of Linnaeus, these entities are no longer terminological “prisoners” in the sense that their variations are not negated; rather, concepts and names are challenged and adjusted according to advances in science.

In summary, from a philosophical and terminological perspective, this “pattern”, as currently worded, is somewhat flawed. This is not to say that the intention behind it is incorrect. Living systems do indeed sense and respond to both their internal environment (integration) and external environment; organisms do communicate in a multitude of different ways with each other and other species, allowing reproduction and survival. The type of perception and communication depends on the species in question (and their associated sensing characteristics). While biologists may not describe this sensing and communication as the acquisition and transmission of “information” it is indeed an essential element of the living world, without which biological systems could not function. These capacities have already inspired biomimetic innovation and will no doubt continue to do so.

**Pattern 6 could be reformulated as:** Living systems sense and respond to their internal/external environments and communicate in a multitude of different ways (physical, chemical, and behavioural).


**Pattern 7. “Nature uses chemistry and materials that are safe for living beings”**


“*Organisms do chemistry within and near their own cells. This makes it imperative that organisms use chemicals, chemical processes, and chemistry-derived materials that are supportive to life’s processes. Life’s chemistry is water-based and uses a subset of chemical elements configured into precise 3D structures. The combination of 3D architecture and composition is the key to maximising self-assembly, guiding chemical activity and material performance, and allowing for biodegradation into useful constituents when their work is done. With regard to our production systems, the importance of using life-friendly chemistry and materials is applicable at various system scales, from sourcing or growing of materials, to manufacturing products or goods, transporting those goods, and considering what happens to them at the end of their life cycle*”.

There is some truth in this “pattern”, though once again it is context-dependent. Whether chemistry and materials are “safe” or “dangerous” depends on the species in question, their life history stage, their environmental context, and, last but not least, the quantity of the chemical compound in question. Taking this “unifying pattern” at face value implies one could eat any amount of anything growing in the wild under the assumption that it would be “supportive to life’s processes”!

Plants synthesise or concentrate hundreds of toxic (sometimes lethal) compounds to protect themselves from other plants or animal predators. For instance, 10,000 to 12,000 plant species are recorded as being toxic to humans. Some insects even ingest plant toxins (pyrrolizidine alkaloids) and store them within their own bodies to protect themselves from being eaten [50]. Evidently, what is safe for one species can be deadly for others. Additionally, it is not just plants that create toxins: many animals also synthesise toxic compounds. Such “biotoxins” are produced by some frogs, insects, and cone snails, as well as venomous snakes, spiders, and jellyfish. Even chemicals that are now fully integrated into our human physiology and considered harmless have not always been so during the course of evolution. For example, steroid hormones have their origins in ancient detoxification pathways of xenobiotics (chemical substances not naturally produced or expected to be present within an organism). These xenobiotics were not “safe” for humans, at least not originally. However, once degraded into cholesterol metabolites, these compounds were subsequently recruited for hormonal signalling functions between cells via a process known as “molecular domestication” [51,52].

The amount of a given substance is also relevant. Humans need oxygen, but above a certain concentration, it becomes neurotoxic. If we were transported back to the carboniferous period with its 0_2_ levels at 30%, we would suffer serious headaches! The same applies to many compounds vital for human health in small amounts (such as Vitamin A), yet dangerous in excess.

With these caveats in mind, what can be usefully retained from this “pattern” for a bioinspired designer? At a macroscale, given enough time and appropriate conditions, almost all organic matter can indeed be broken down by microorganisms (such as bacteria and fungi) into CO_2_, water, and minerals. That is to say, organic matter biodegrades into “safe” constituents under the right circumstances (see Pattern 2). Human beings, on the other hand, have created materials that act in opposition to “life’s processes” over longer time scales. Whilst modern man-made non-organic material can also biodegrade, this often takes much longer: an aluminium can, for example, is estimated to take 100–200 years to biodegrade, compared to 1 month for a vegetable. At current rates, this anthropogenic waste is accumulating faster than it can biodegrade, threatening biodiversity’s ability to provide vital ecosystem services. More troubling are synthetic compounds known as Persistent Organic Pollutants (or “forever chemicals”) that resist biodegradation and bioaccumulate in the food chain, reaching dangerous levels of concentration (e.g., Dichlorodiphenyltrichloroethane known as DDT, Chlordecone). The bioinspired designer could seek to develop innovative materials that biodegrade over timeframes in line with human lifespans (i.e., decades rather than centuries) and/or those that can be efficiently recycled without wider chemical contamination.

**Pattern 7 could be reformulated as:** Whether the chemicals and materials synthesised within biological systems are “safe” depends on the species in question, their life history stage, their environmental context, and, last but not least, the quantity of the chemical compound in question. Nevertheless, almost all are ultimately biodegradable, given sufficient time and the right environmental conditions.


**Pattern 8. “Nature builds using abundant resources, incorporating rare resources only sparingly”**


“*Nature’s materials are abundant and locally sourced. This is true whether an organism is building something external to itself, like a termite mound or a nest, or assembling materials that are part of the body, e.g., a wing, shell, leaf, or horn. The most common and abundant basic building blocks—chemical compounds—are those that are formed from the most common and readily found elements on earth: carbon, nitrogen, hydrogen, and oxygen. A few rarer minerals are also used, but these are found locally and are readily available, not mined, processed, or shipped thousands of miles. Waste is eliminated through additive manufacturing and by building processes around readily available and low-cost sources of materials and energy*”.

There is much to agree with in this “pattern”: living systems do indeed build materials principally from abundant elements. Calcifying organisms, such as echinoderms, molluscs, and corals, produce shells and skeletons composed principally of calcium carbonate. Similarly, diatoms, a group of microalgae that constitute almost half of all oceanic organic material, have cell walls made of silica. It is true to say that these minerals are locally abundant.

However, this pattern’s wording reveals a circular argument: if a resource is rare and if “nature” is considered to be all organic matter (at different levels of organisation) [7], then “nature” is *inevitably* obliged to use that resource sparingly. It is simply not possible to incorporate a great deal of a rare resource (at least not without human technology). At the scale of an organism, if a certain resource is required but not readily available, the organism is not able to survive.

With regard to waste, not all is “eliminated”. Living systems (over their 3.7 billion years of evolution) have had episodes of producing “waste” that could not be assimilated locally, leading to death and extinctions (see Pattern 2. “Nature recycles all materials”). At a species level, some organisms accumulate minerals that appear to be “wasted” when they die. For example, tunicates concentrate rare metals, such as Vanadium [53], in their tissues, potentially to protect themselves from predators. This accumulation is “lost” back to the surrounding environment at the end of the tunicate’s life. Equally, “wasting” resources is common among carnivores: wolves, foxes, weasels, and martens all kill more prey than they are able to eat when prey is abundant and accessible. As with the notion of “frugality”, “waste” is essentially a human concept that has little meaning in the context of biological systems.

Once again, our explanations are not intended to imply that frugality is an invalid concept in the context of sustainable design. Using abundant, locally sourced resources in an efficient way and minimising waste are all excellent principles to follow in the pursuit of environmentally friendly innovation.

**Pattern 8 could be reformulated as:** Most biological materials are inevitably composed of abundant, locally available resources.


**Pattern 9. “Nature is locally attuned and responsive”**


“*Chances of survival increase when individuals are good at recognizing local conditions and opportunities and locating and managing available resources. Survival also depends on responding appropriately to information garnered from the local environment. Organisms and ecosystems that are present in a location evolved in direct response to local environmental conditions. Some of those environmental conditions change in a cyclic pattern, such as tides, day and night, seasons, and annual floods or fires. Organisms use those predictable cyclic patterns as an opportunity, evolving to fill a particular niche. Within a particular location, there are micro-environments, such as a low spot that is moister than the surrounding area or an area that experiences more wind than others. These also provide opportunities for organisms to have an advantage over others and thrive. Some environmental conditions change slowly over time as the climate changes or as the organisms and ecosystems influence the local conditions. Being able to respond to these changes, again using them as opportunities, allows organisms and ecosystems to flourish*”.

Living systems certainly respond to their surrounding environment, both biotic and abiotic. However, this “pattern” appears to confuse three different processes: responsiveness, acclimatisation (both at an individual scale), and adaptation (at a population scale).

*Responsiveness* is an organism’s capacity to adjust to internal and external stimuli. For example, animals move away from fire; many plants turn towards the sun; and single-celled organisms may migrate towards a nutrient source or away from a harmful chemical. Responsiveness is a trait common to all living organisms because those with lower or no responsiveness died.

*Acclimatisation* (also called acclimation or acclimatation) refers to changes that occur during an individual organism’s lifetime to improve its survival and/or reproduction in response to new environmental conditions. These changes can be *transient and reversible* and relate to an organism’s behaviour, morphology, or physiology. For example, some mammals grow thicker fur in the winter to protect them from the cold.

*Adaptation* involves changes (behavioural, morphological, or physiological) that occur over many generations at the level of populations. It is the way in which a population of organisms *stabilises* a trait under a given set of environmental conditions. Adaptation usually refers to hereditary traits that have evolved through natural selection. An example of adaptation would be a fire-tolerant plant that has evolved to live in a fire-prone Mediterranean climate.

While *individual* organisms can be responsive and able to acclimatise, they are ultimately steered by stronger evolutionary constraints at a *population* scale: certain traits are selectively filtered over each generation. Overall, adaptation is the result of (i) environmental conditions, (ii) advantageous/disadvantageous hereditary traits that exist within a population, and (iii) historical constraints (phylogenetic heritage). Adaptation at the population scale is the byproduct of an “invisible cemetery” from an evolutionary point of view; it is not a possibility that can be pursued at the level of an individual organism.

Nevertheless, much can be learned from individual living systems’ ability to respond and acclimatise to new conditions. Indeed, the capacity to respond to change and identify unexplored niches in different fields of innovation is essential for a bioinspired designer.

**Pattern 9 could be reformulated as:** Individual organisms are responsive and often able to acclimatise to new environmental conditions. At the population level, organisms continually adapt to their surroundings through natural selection.


**Pattern 10. “Nature uses shape to determine functionality”**


“*Nature uses shape or form, rather than added material and energy, to meet functional requirements. This allows the organism to accomplish what it needs to do using a minimum of resources. Forms can be found in the shape of a beetle’s back and in the multi-layer structure of a tropical rainforest. If we notice a form in nature, with very rare exceptions, there’s almost always a functional reason behind that form*”.

It is accurate to say that structural complexity (“shape”) is behind the vast array of multi-functional biological materials found in nature. Most biological materials are composed principally of a limited number of elements that are brought together to form a few base compounds (mainly nucleic acids, proteins, lipids, polysaccharides, and minerals) [54]. With limited chemical variation and minimal energy, organisms create a huge range of materials by way of complex, hierarchical structures (at nano and microscales). It is this *microstructural diversity* that underpins the very different properties of biological materials and their multi-functionality. For example, compare the flexible suckers of an octopus with the hydrophobic carapace of the Namib Desert beetle or even the iridescent wings of blue Morpho butterflies: it is the *structural* properties of their biological materials (rather than their *chemical* complexity) that deliver their very different functionality. These nano/microstructural properties are of great interest to bioinspired material engineers.

However, care must be taken in how we attribute living systems’ functionality to their shape. As already discussed, humans use form to determine functionality because we consciously anticipate the purpose of the tool we need and then shape it accordingly. In nature, it is functionality that determines form; function is the operative condition via which certain forms thrive (or disappear) in response to environmental conditions. A purposeless form that appeared at random through genetic mutation is passed down to following generations only if that form offers some (even slightly) advantageous functionality. Where there is no useful function, a particular form will disappear. In caves, for example, visual functions are useless. This means that having a nervous system responsive to photons (light) is no longer relevant to survival. As a consequence, random alterations within visual systems cease to be disadvantageous. These changes accumulate over generations until eyes are lost entirely. This explains why cave ecosystems contain eyeless crayfishes (e.g., *Orconectes australis*), eyeless spiders (e.g., *Adelocosa anops*), eyeless fishes (e.g., *Stygichthys typhlops*, *Astyanax mexicanus*, *Amblyopsis hoosieri*, *Satan eurystomus*), eyeless amphibians (e.g., *Proteus anguinus*), several distinct lineages of eyeless amphipods (*Niphargus* spp.), etc.

As such, the last sentence in this pattern’s explanation is the most accurate: “*If we notice a form in nature, with very rare exceptions, there’s almost always a functional reason behind that form*”. In other words, functionality determines form. This principle can be directly applied to bioinspired design, even though the process of human creation is inherently different from biological adaptation.

A final point: this pattern mentions the “*multi-layer structure of a tropical rainforest*” as an example “form” within nature. One might conclude that the “function” of this multi-layer structure is to allow many different species to flourish and interact based on their relative requirements for water, sunlight, air circulation, etc. However, at an ecosystem level, the relationship between form and function is much more complex, involving a much greater range of factors. Ecosystem functions include flows of materials and energy, biogeochemical cycling, and relationships among organisms and between organisms and their environment. The ecosystem structures that support these functions are varied and interconnected; absolute relationships between a specific form and function are therefore more difficult to establish.

**Pattern 10 could be reformulated as:** In biological entities, functionality determines form. Structural complexity, rather than chemical composition, is behind the vast array of multi-functional biological materials found in the natural world.

## 4. Eight Biological Principles for Enlightened Bioinspiration

Over and above our suggested revision of the Biomimicry Institute’s ten “unifying patterns of nature”, we would like to suggest eight further biological principles for consideration by bioinspired designers (Table 2). Some are intuitive; others depend on an understanding of how biological systems have developed and evolved through time. We focus on the latter in this paper, as these principles are often overlooked or misunderstood. In addition, we concentrate on biological principles applicable at the organismal level. This means we place less emphasis on levels lower than the cellular scale or on higher levels of ecological organisation (such as populations, guilds, ecosystems, biomes, or the biosphere). These are all “living systems” but not living organisms, and therefore different “principles” may apply.

### 4.1. Living Organisms Are Discrete (i.e., Bounded), Self-Organised, Self-Maintained, Thermodynamically Open Systems That Modify Their Surrounding Environment

Organisms have a physical embodiment, with boundaries defining an inside and an outside. Inside this boundary, organisms are self-organised and self-maintained (or “autopoietic”). The term “autopoiesis” comes from the words “auto-” meaning “itself” and “-poiesis” meaning “creation”. Autopoiesis is based on the theory that living systems consist of a network of processes (production, transformation, and destruction) that create and regenerate themselves [55]. More specifically, the following qualities have been attributed to organisms via the concept of autopoiesis [56]:

*Self-development (and self-maintenance)*: aside from obtaining nutrients and expelling waste, living systems autonomously maintain and restore their own structural integrity insofar as is possible (excluding definitive damage, collapse, and death).

*Emergence and non-localisation*: living organisms are not defined by their units (proteins, cells, organs, etc.) but rather by the properties that emerge from *interactions* between these units at every level of biological complexity.

*Interaction with the environment*: Boundaries (such as a cell membrane or skin) must allow exchanges of matter/energy/sensation in the form of inputs, allowing internal metabolic processes to take place. They must also allow outputs into the supersystems (or environment) that host these organisms (in the form of waste, behaviour, communication, etc.).

Organisms are thermodynamically open (i.e., “dissipative”), meaning that they are in non-equilibrium and constantly exchanging matter and energy with their surrounding environment [57]. Indeed, the self-organisation found in organisms is a *consequence* of their ability to export entropy (waste, heat, etc.) into their ecosystem. That is to say, living systems create structure at their individual scale by “getting rid of” energy/matter in various forms. In doing so, the wider environment is modified. The more complex the internal *organisation*, the greater the organism’s *outputs*, and the more significant the external *alteration*.

At the ecosystem level, mechanisms have emerged for mitigating against these destructive “entropic effects” generated by organisms. These include circular flows of matter and energy, their redistribution via food webs, the use of elements easily (re)incorporated into living organisms (e.g., carbon, hydrogen, nitrogen, and oxygen), and functional diversity providing ecosystem resilience.

What relevance does this have to bioinspiration? Bioinspired designers need to understand that their “biological muses” are the result of self-organised processes connected to their wider environment (being changed by it and changing it in return). Taking an organism out of context in the design process can affect the success of the product or process it inspired. Designers also need to reflect on the “optimal rate” of entropy in order to ensure a bioinspired innovation is as sustainable as possible in a given environment.

### 4.2. Living Organisms Are Unique; They Vary at Random; Apparent Biological “Order” or Regularity at the Population Scale Is a Consequence, Not a Cause

Looking at the natural world, a designer may be struck by certain regularities, common forms, and the appearance of organisation. Indeed, biological systems *are* organised in the sense that they contain specialised, coordinated, and nested parts, and organisms are made up of cells (the fundamental units of life) in autopoeitic systems.

These observations may lead the designer to think that living systems are governed by laws, such as in physics. Yet, while all entities are bound by physical laws, physics alone cannot explain biological systems. This is because biology is a science underpinned by *random variation* (of genes, organisms, and habitats) and *history* (see Principle 3 below). While physics works with *universals* (e.g., atoms, mass, and energy) and *laws*, biology works with *particulars* (that tree, your cat, etc.) and *principles*. Biology seeks to explain, through these principles*,* how biological organisation arises from unstructured phenomena at a range of different scales.

One of biology’s guiding principles is natural selection [58]. In his study of this principle, Charles Darwin paid much attention to organisms’ variation and its consequences at the population level. He wanted to understand why individuals of the same species were similar when there was so much initial variation among organisms that interbred. Natural selection, in the short term, provided an explanation: each generation of organisms is “pruned” and extreme variants eliminated. Observing the resultant similarity between the remaining individuals, the concept of “species” was developed (as a terminological convention rather than a biological reality) [59]. For Darwin, natural selection acted principally as a stabilising factor. This is reflected in the subtitle of his most well-known book: “*The origin of species by means of natural selection or the preservation of favoured races in the struggle for life*”. The words “evolution”, “transformation” or “transmutation” are notably absent. Rather, the word “preservation” is used to convey the maintenance of a short-term status quo.

Obviously, if an environment changes over time, the average form of a species within that environment will change in response. Nevertheless, natural selection contributes to the apparent stability, similarity, or regularity across individual organisms of the same species. Darwin’s genius was to demonstrate, for the first time since Maupertuis in 1751 [16], how biological structure and function can appear from “purposeless” random variation in the living world. Later, other biologists explained how structures/regularities arose at other biological scales, for example, through changes in chemical states, stochastic variations in genetic expression [60,61,62], and variation among cells of the same organism [62,63,64,65,66,67,68,69].

### 4.3. Living Organisms Are Shaped by Their Phylogenetic Heritage

Natural selection does not operate in a vacuum to produce “perfectly adapted” organisms ready for the bioinspired designer to imitate; rather, it has to work within historical constraints associated with species’ evolutionary trajectories [70,71,72]. Organisms retain traces of changes in their lineages, a property called “biological historicity”. Indeed, individual organisms can be defined as “integrated entities complex enough to keep track of their historical trajectory” [73]. Consequently, evolutionary history offers fundamental explanations as to why organisms are as they are.

For example, if one were to design an organism well adapted to childbirth, it would not be *Homo sapiens*. Our bipedal skeleton (with a forward-tilting pelvis) and our comparatively large skulls make the exercise painful, even deadly. This arrangement is the result of historical constraints: when our hominoid ancestors started walking upright (restricting their pelvic structures), head size was not an issue. However, some two million years ago, *Homo* brains tripled in volume. Meanwhile, birth canals did not change significantly, leaving them inherently “unadapted” to birthing large-brained offspring. Functional explanations from physics and physiology are insufficient to explain why a human being is shaped as it is. Evolutionary history is also needed.

Similarly, the “strange” structure of the heart’s aortic arch, the “absurd” connection between the optic nerve and retina in vertebrates, or the three seemingly useless muscles of humans’ outer ear: these peculiarities can only be explained as a consequence of organisms existing “downstream” of a given evolutionary line. The same applies to certain responses; for example, humans getting “goose bumps” when cold is simply a remnant of our mammalian fur function that is of no use today; equally, the grasping reflex of a newborn baby is a useless leftover from our primate past. These traits were certainly not “designed” with efficiency in mind; it is only phylogenetic heritage that accounts for them [74]. An organism’s “historicity” should be understood before biological models are transposed to human innovation.

A note of caution: sometimes very similar structures are found in two distinct phylogenetic groups. One example is the hydrodynamic form characteristic of both sharks and dolphins. This phenomenon is confusingly referred to as “convergent evolution”, but this term is misleading. Comparable physical, chemical, and environmental constraints have, through natural selection, shaped two distinct lineages of organisms independently of each other in a way that appears similar to us at our scale of observation. However, all living organisms being unique, evolution is “convergent” only in the fact that we name these structures the same way. There is no specific evolutionary process beyond this.

### 4.4. Living Organisms Metabolise to Grow and Achieve Homeostasis

Living organisms undergo regulated growth. Individual cells grow larger, and muticellular organisms accumulate many cells through cell division. Life also depends on a large number of interconnected chemical reactions inside these cells, known collectively as metabolism. Metabolism enables organisms to move or catch prey, to convert energy from one form to another, and to grow, reproduce, and maintain their physical structure. Metabolism functions because organisms regulate their internal environment to maintain a narrow range of conditions, even in the face of external change. This is called homeostasis.

Why are these concepts of growth, metabolism, and homeostasis relevant to a bioinspired designer? Firstly, because these “life traits” can inspire new products and processes outside the traditional engineering/design frameworks. For example, plant growth (and changes that occur during the growth process) have inspired dynamic biomimetic building construction [75]. Similarly, the principles of homeostasis can be applied to urban design processes [76] and even the concept of metabolism to architecture [77].

However, perhaps more importantly, these characteristics determine how designers can appropriate biological “solutions” and apply them to their technical challenges. On the one hand, artificial structures do not have to achieve homeostasis, metabolise, or grow, thus “freeing” the design process from these biological constraints. On the other hand, some biological systems only succeed *because* they are alive; their living traits cannot be entirely transposed to technology (see also “autopoiesis” under Principle 1). In this context, designers may seek to integrate biological entities directly into the design process (as is the case in biodesign/bioassistance) rather than trying to create artificial replicas.

### 4.5. Living Organisms Multiply, and All Species Have the Potential to Proliferate

Another basic principle of life, as observed by Darwin, is reproduction. Living things multiply to create new organisms. Reproduction can be either asexual, involving a single organism, or sexual, requiring two parents. Single-celled organisms reproduce simply by dividing into two. The genealogical lineages of organisms today exist because their ancestors directed energy towards producing offspring. At the species level, each generation produces more offspring than the local resources can support. Those that did not went extinct.

Reproduction is curtailed by certain external limits, both biotic (for example, predation, and food availability) and abiotic (for example, temperature, pressure, and salinity). These limits are precisely why we rarely see sudden proliferations of individual species within natural, “healthy” ecosystems. It is only when this balance is disrupted, for example, when invasive species are introduced into new habitats, that such extreme population expansion occurs (see revised “Pattern 4”). This continues until new complex equilibria among species are eventually established (which may take centuries). The take-home message for bioinspired designers is that certain limits structure all populations through selective constraints. There is no “unlimited” world where unfettered growth (human or otherwise) can continue indefinitely.

### 4.6. Living Organisms Transmit through Both Genetic and Non-Genetic Processes

One basic property of biological entities is their capacity for transmission at all scales of organisation. Genetic transmission is well known: bacteria transmit their genetic variations through division; somatic cells transmit their genetic material through mitosis (asexual cell division); and through meiosis and fertilisation, most multicellular organisms “mix” male and female genes for transmission to the next generation (sexual reproduction).

However, the term transmission should not imply that genetic material is simply a set of “instructions” to be followed (as the misleading and now obsolete “genetic program” metaphor suggests) [11,14,23,78]. In modern biology, the deterministic role of genes has been actively reexamined [12,23,24,79,80,81,82]. Scientists have concluded that, unlike following a blueprint to design an object, genetic and non-genetic transmission are influenced by many factors, including an organism’s (i) biological development, (ii) life traits [83], (iii) wider population, and (iv) surrounding environment. Genes do not “control”, they simply provide an impulse; similarly, development is no longer understood as the deployment of a “programme” but as a construction process that participates in the reproduction of the phenotypic trait itself [24].

Moreover, transmission is not limited to genes. For example, at the nanoscale, normal prion proteins’ tridimensional shape can be altered via contact with abnormal, pathogenic prions (causing neurodegenerative disorders). Organisms can also transmit epigenetic states, egg cytoplasm, symbionts, behaviours, techniques, cultures, and ecological niches (“inclusive inheritance” [24,84,85]) from one generation to the next. One might even consider bioinspired design to be the transmission of biological “ideas” from one living system to another!

### 4.7. Living Organisms Die, Leading to Better Adaptation and Long-Term Survival of Their Lineage

Immortality is not a biological property, as much as it may intrigue humanity. Although it could seem counterintuitive, death is essential for the continuation of a biological lineage in the face of environmental change. This is because death allows individual organisms to be replaced within a population. This makes way for other individuals with different traits, as well as “rearranging” variations within a population. It is this continual influx of variation that allows a lineage to adapt to unpredictable abiotic and biotic challenges.

Death is therefore a part of life, a precursor to the variation biological systems need to adapt and survive in the long term. Human beings are not accustomed to this concept of mortality, which we tend to experience as an individual-centred occurrence. This presents a challenge for bioinspiration with its design-oriented optimisation approach. It can be difficult for a designer to understand that the process that gave birth to “interesting design solutions” occurred purposelessly at a population level simply due to random variations and a great deal of death. It is only through acknowledging the mortality of individuals that did not succeed that we can explain the apparent “optimisation” of the living world without the role of an omnipotent designer.

### 4.8. Living Organisms Evolve under Constraints

Variation, transmission, and environmental constraints lead to the phenomenon of natural selection (including sexual selection) and then adaptation. As we have seen, this mechanism is at the origin of biological *functionality* and the apparent congruence between function and form. This process also leads to similarity among individuals of the same species.

Environmental constraints (both biotic and abiotic) drive natural selection (and, to a certain extent, organisms do as well [24]). We can control environmental change experimentally, for example, by choosing constraints that act on a population of short generation organisms like bacteria or yeast. This method of mimicking natural selection is used in protein engineering, where a gene is submitted to iterative rounds of mutagenesis and selective constraints. This process is called “directed evolution” and its “metabolic engineering” allows the production of compounds or new metabolic pathways useful to industry by using millions of random mutations to obtain a pre-defined result [86,87]. Such “domestication” of natural processes in laboratories could be considered a type of bioinspiration, inspired by natural selection and evolution itself.

Why is this eighth principle important? Without an understanding of evolution, a bioinspired designer might isolate a certain anatomical structure and seek to transpose it to technology, ignoring the constraints that drove the structure’s existence. They might try to replicate an optimisation that does not actually exist or fail to integrate parts of the structure that are essential for its operation.

For example, the ovipositor of the wood wasp *Megarhyssa nortoni nortoni* is essentially a thin, stiff, and strong drill. It can withstand high forces and penetrate wood in a curved fashion. These traits inspired the design of an innovative intracranial endoscope [88,89]. The artificial drill was initially made of a soft polymer to allow the surrounding cranial tissue to react to the side forces generated. However, the softness of this polymer compromised the stability of the mechanism that linked the constituent parts of the drill together. In other words, an essential part of the ovipositor’s functioning was initially overlooked because it was not an obvious human design constraint. Ultimately, it became necessary to reconsider the trade-offs associated with the drill’s function using multi-objective analyses and modelling in order to achieve a workable solution.

Removing the biological “solution” from its broader biotic and abiotic (environmental) context may be inevitable in design, but it should always be pursued with awareness of the evolutionary context, difficult though that may be. This is challenging because, along the evolutionary pathway that led to a specific organism in the here and now, previous generations will have responded to conditions that we cannot, *a priori*, fully apprehend or understand simply by analysing today’s environment. In this sense, palaeontology could become a powerful ally in improving bioinspired design.

## 5. Conclusions

The way biologists understand life is diametrically opposed to the way designers and engineers create. This is why teaching evolution correctly is challenging [90]. Life is functional not because of a design but due to filtered random variation, with failures outnumbering successes. Death is the rule, and survival the exception. In human design, function is anticipated, and this significantly limits the number of trials and errors. Moreover, almost every biological structure is multifunctional. Some functions may be more important than others, but by and large, living systems are the result of multi-factor trade-offs [70,71,72,90,91]. In the natural world, there are no optimisations, whereas in human design, optimisation is the goal.

Nonetheless, being inspired by the natural world’s forms, processes, and systems is an effective way to innovate sustainably. It is also a compelling means to get people interested in biology, biodiversity, and its preservation. The fact that biological principles are different from the way humans design need not be an obstacle to bioinspiration. However, if bioinspiration is to be practised lucidly, scientific explanations of why structures or processes exist in the natural world must be prioritised. We understand that metaphors may be useful intellectual stepping stones towards more scientifically complex concepts, and we appreciate that this is the likely intention of the Biomimicry Institute’s “Ten Unifying Patterns of Nature”. Nonetheless, we need to be wary of misleading bioinspiration professionals (and the public) with ambiguous statements about “Nature”.

The overriding message of this paper is that bioinspiration does not require the personification of “Nature” or the projection of our own cognitive processes onto a “reified” vision of the living world. We hope our suggested revisions of life’s “Patterns” retain their ability to communicate and inspire without perpetuating inaccurate notions about life on earth. We believe that understanding biological principles and working with biologists will make for better bioinspiration practices; an “enlightened” bioinspiration that respects the theory of biology will be a valuable long-term investment in the field.

## Figures and Tables

**Table 1 biomimetics-08-00362-t001:** Proposed scientific reformulations of the Biomimicry Institute’s ten “Unifying patterns of nature” for an enlightened bioinspiration.

Biomimicry Institute’s Ten “Unifying Patterns of Nature”	Proposed Scientific Reformulation
1. Nature uses only the energy it needs and relies on freely available energy	In terms of reproduction, both internal and external constraints can lead to higher energy expenditure than might be inferred from observation of adult populations. Evolutionary processes require considerable amounts of energy—either from the organism’s point of view (number of gametes produced) or from the population’s (number of deaths). However, natural selection seems to ultimately favour physiological systems that minimise energy expenditure.
2. Nature recycles all materials	The living world has an extraordinary (but not infallible) capacity to recycle organic material. In any given ecosystem, a diversity of organisms reuse, scavenge, or decompose matter into components taken up by other forms of life. However, “recycling” can take millions of years, and some organic materials have never been “recycled” at all.
3. Nature is resilient to disturbances	Ecosystems and biological entities are resilient to disturbances only within certain limits. At the ecosystem level, once certain disturbance thresholds are crossed, the “identity” of the ecosystem may be changed irreversibly.
4. Nature tends to optimise rather than maximise	Living systems are the result of trade-offs, not optimisation. Populations seem to ‘maximise’ reproduction and offspring, which are later filtered by environmental constraints (biotic and abiotic). Apparent optimisations in terms of species’ physical and behavioural traits would be more accurately described as being the ‘best under the circumstances’.
5. Nature provides mutual benefits	Mutually beneficial relationships are found in living systems, yet they are not necessarily more significant than predation and parasitism.
6. Nature runs on information	Living systems sense and respond to their internal/external environments and communicate in a multitude of different ways (physical, chemical, and behavioural).
7. Nature uses chemistry and materials that are safe for living beings	Whether the chemicals and materials synthesised within biological systems are “safe” depends on the species in question, their life history stage, their environmental context, and, last but not least, the quantity of the chemical compound in question. Nevertheless, almost all are ultimately biodegradable, given sufficient time and the right environmental conditions.
8. Nature builds using abundant resources, incorporating rare resources only sparingly	Most biological materials are inevitably composed of abundant, locally available resources.
9. Nature is locally attuned and responsive	Individual organisms are responsive and often able to acclimatise to new environmental conditions. At the population level, organisms continually adapt to their surroundings through natural selection.
10. Nature uses shape to determine functionality	In biological entities, functionality determines form. Structural complexity, rather than chemical composition, is behind the vast array of multi-functional biological materials found in the natural world.

**Table 2 biomimetics-08-00362-t002:** Eight biological principles for an enlightened bioinspiration.

Biological Principles (at the Organism Scale)	Take-Home Messages for Bioinspired Practitioners
1. Living organisms are discrete (i.e., bounded), self-organised, self-maintained, thermodynamically open systems that modify their surrounding environment	Organisms have a physical embodiment, with boundaries defining an inside and an outside. Inside this boundary, organisms are self-organised and self-maintained. Organisms are also thermodynamically open, constantly exchanging matter and energy with their surroundings. Bioinspired designers should recognise that their “biological muses” are connected to their wider environment (being changed by it and changing it in return).
2. Living organisms are unique; they vary at random; apparent biological “order” or regularity at the population scale is a consequence, not a cause	Looking at the natural world, a designer may be struck by the appearance of organisation. Indeed, biological systems *are* organised in the sense that they contain specialised, coordinated, and nested parts, and that organisms are made up of cells. However, biology cannot be explained by laws. Rather, it is underpinned by random variation and governed by overarching principles such as natural selection. Natural selection contributes to the apparent stability or regularity across individual organisms of the same species.
3. Living organisms are shaped by their phylogenetic heritage	Natural selection does not operate in a vacuum to produce “perfectly adapted” organisms ready for the bioinspired designer to imitate; rather, it has to work within historical constraints associated with species’ evolutionary trajectories. Organisms retain traces of changes in their lineages, a property called “biological historicity”. An organism’s phylogenetic heritage should be understood before its characteristics are transposed to human innovation.
4. Living organisms metabolise to grow and achieve homeostasis	Living organisms undergo regulated growth. Life also depends on a large number of interconnected chemical reactions (metabolism). Metabolism functions because organisms regulate their internal environment, even in the face of external change (homeostasis). These characteristics determine how designers might appropriate biological “solutions” (both in terms of constraints and opportunities).
5. Living organisms multiply, and all species have the potential to proliferate	Living things reproduce (sexually or asexually). At the species level, each generation produces more offspring than the local resources can support. Those that did not went extinct. Reproduction is curtailed by certain external limits, both biotic and abiotic. There is no “unlimited” world where unfettered growth (human or otherwise) can continue indefinitely.
6. Living organisms transmit through both genetic and non-genetic processes	Biological entities have the capacity for transmission at all scales of organisation. Unlike following a blueprint to design an object, genetic transmission is influenced by many factors. Moreover, transmission is not limited to genes; it can also apply to symbionts, behaviours, ecological niches, etc. One might even consider bioinspired design to be the transmission of biological “ideas” from one living system to another!
7. Living organisms die, leading to better adaptation and the long-term survival of their lineage	Death is essential for the continuation of a biological lineage in the face of environmental change. It provides an influx of variation that allows a lineage to adapt to unpredictable abiotic and biotic challenges. Only by acknowledging the mortality of individuals that did not succeed can we explain the apparent “optimisation” of the living world without the role of an omnipotent designer.
8. Living organisms evolve under constraints	Variation, transmission, and environmental constraints lead to the phenomenon of natural selection (including sexual selection) and then adaptation. Without an understanding of evolution, a bioinspired designer might isolate a certain anatomical structure and seek to transpose it to technology, ignoring the constraints that drove the structure’s existence.

## Data Availability

Not applicable.

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
