# Peer review of "Revisiting Nature’s “Unifying Patterns”: A Biological Appraisal"

_biomimetics, 2023, doi:10.3390/biomimetics8040362_

Round 1

Reviewer 1 Report

The paper by Lecointre et al. entitled “Revisiting Nature’s “unifying patterns”: a biological appraisal” presents a long discussion on nature’s unifying patterns. The manuscript style differs from the typical research manuscript but it is very well written and raises topics that can be helpful for the biomimicry community, ranging from engineers to biologists. A few points are raised:  

·         Are the authors affiliated with the Institute? Also, a brief introduction to the Biomimicry Institute would be enhance the manuscript.

·         Page 2, number 4. In the left column, it is stated that Nature optimizes; while in the right, that it does not. Please clarify it.

·         Please adjust English following either the British or American spelling. For example, on page 2, the manuscript uses: “optimize” and also “optimisation”.

·         What is the difference between items 4 and 9?

·         The discussion on page 3 is very interesting. Following the statements in lines 85-107, maybe the paper below can be helpful. In this paper, they question the usage of “optimally designed” biological materials.

o   https://doi.org/10.1098/rsif.2022.0336

·         Citation 50 is out of format.

Please adjust English following either the British or American spelling. For example, on page 2, the manuscript uses: “optimize” and also “optimisation”.

Author Response

Thank You. Changes have been made.

Reviewer 2 Report

This is a VERY excellent review of “natural principles” and their uses for technics, but not limited to that. The authors have a deep understanding of biology, evolution and engineering. It is one of the reviews that I most enjoyed reading. It will be a very fine contribution to “Biomimetics”

I have some minor issues/comments:

Please rephrase the first sentences of the introduction (lines 29-34) which are a 1:1 copy of the abstract.

Page 2, bottom as well as page 14, line763: missing word:  “composed of” or “composed from”

Page 4, line 97: Missing word: “in order to generate”

Line 218: There might be two spaces before “This…”

This might be a general (albeit minor) issue, i.e “one space too many”. Lines 328, 345, 359 and many other instances throughout the text and in the references between “[xx]:” and the first author’s name. Please check carefully

Pages 10: lines 239-430: I fully agree with the authors that state that “Species become extinct: the world has seen five big extinction events prior to the Anthropocene.” However, a citiation would be well placed here to strengthen this statement.

Line 511: There are debatable examples for a species being perfectly adapted in relation to a single function. Like the human eye being as sensitive as to detect one photon.

Line 610: “are indeed central elements”

Lines 980, 981, 987 and other instances in the text: [60-62] instead of [60, 61, 62]; [62-69] instead of [62, 63, 64, 65, 66, 67, 68, 69], etc. ?

Lines 1211-1212: please issue abbreviation for journal Acta Palaeontologica Polonica

Lines 1220-1221 and many others: please always use OR always do not use abbreviations for journals (depending on format requirements), please check carefully. As well as please always diplay DOI information(where available) OR always do not display DOI information-

Line 1280, 1302 and others: please change format of DOI to the journal’s format!

Author Response

Thank You. Changes have been made.
